# Everything, Everywhere, All at Once: Is Mechanistic Interpretability Identifiable?

**Maxime Méloux, François Portet, Silviu Maniu, Maxime Peyrard**
Université Grenoble Alpes, CNRS, Grenoble INP, LIG, 38000 Grenoble, France
`{melouxm,portetf,manius,peyrardm}@univ-grenoble-alpes.fr`

## Abstract

As AI systems are increasingly deployed in high-stakes applications, ensuring their interpretability is essential. Mechanistic Interpretability (MI) aims to reverse-engineer neural networks by extracting human-understandable algorithms embedded within their structures to explain their behavior. This work systematically examines a fundamental question: for a fixed behavior to explain, and under the criteria that MI sets for itself, are we guaranteed a unique explanation? Drawing an analogy with the concept of *identifiability* in statistics, which ensures the uniqueness of parameters inferred from data under specific modeling assumptions, we speak about the *identifiability of explanations* produced by MI. We identify two broad strategies to produce MI explanations: (i) "where-then-what", which first detects a subset of the network (a circuit) that replicates the model's behavior before deriving its interpretation, and (ii) "what-then-where", which begins with candidate explanatory algorithms and searches in the activation subspaces of the neural model where the candidate algorithm may be implemented, relying on notions of causal alignment between the states of the candidate algorithm and the neural network. We systematically test the identifiability of both strategies using simple tasks (learning Boolean functions) and multi-layer perceptrons small enough to allow a complete enumeration of candidate explanations. Our experiments reveal overwhelming evidence of non-identifiability in all cases: multiple circuits can replicate model behavior, multiple interpretations can exist for a circuit, several algorithms can be causally aligned with the neural network, and a single algorithm can be causally aligned with different subspaces of the network. We discuss whether the unicity intuition is necessary. One could adopt a pragmatic stance, requiring explanations only to meet predictive and/or manipulability standards. However, if unicity is considered essential, e.g., to provide a sense of understanding, we also discuss less permissive criteria. Finally, we also refer to the inner interpretability framework that demands explanations to be validated by multiple complementary criteria. This work aims to contribute constructively to the ongoing effort to formalize what we expect from explanations in AI.

## 1 Introduction

Interpretability in machine learning spans diverse goals and methods (Molnar, 2022; Carvalho et al., 2019; Teney et al., 2022), from creating inherently interpretable models to applying post hoc techniques to explain model decisions. *Mechanistic interpretability* (MI) aims to reverse-engineer models to reveal simple, human-interpretable algorithms embedded in neural network structure (Olah et al., 2020). MI is focused on generating what we call *computational abstractions*, where complex neural networks' behaviors are explained by simpler algorithms that track the internal computations (Olah et al., 2020). A computational abstraction – a mechanistic explanation – has two components: (a) *what* is the explanatory algorithm, and (b) *where* in the computational structure is this algorithm embedded? Given the intractability of exhaustively searching all possible algorithms across all subsets of a neural network, researchers have developed methods with different assumptions and trade-offs. We categorize these methods into two broad strategies. The first, which we call *where-then-what*, focuses on finding a subset of the network – a *circuit* – that captures most of the information flow from inputs to outputs. Once this circuit is identified, typically using heuristics,

the next step is to interpret its components (*features*) to derive the explanatory algorithm (Dunefsky et al., 2024; Davies & Khakzar, 2024; Conmy et al., 2023). The second approach, which we name *what-then-where*, starts by identifying candidate algorithms and then searches subspaces in the neural network where the algorithm may be implemented. This is performed using causal alignment between the explanatory algorithm's states and the network's internal states and typically requires approximation algorithms (Geiger et al., 2022a;b). Each strategy relies on specific criteria to assess candidate explanations. For instance, circuits can be evaluated by their *circuit error*, which quantifies how closely the circuit's predictions match the full model ones (Conmy et al., 2023). In the *what-then-where* strategy, candidate algorithms are compared based on causal alignment measures like intervention interchange accuracy (IIA), which assesses how well the algorithm's states remain aligned with the network's internal states after counterfactual manipulations of the states.

In this work, we question a property of explanation that appears to be tacitly taken for granted: do MI criteria guarantee a unique explanation of a fixed behavior? The concept of identifiability is well-established in statistics, where a model is identifiable if its parameters can be uniquely inferred from data under a given set of modeling assumptions (e.g., Rothenberg, 1971). By analogy, we extend this terminology to interpretability, defining the identifiability of explanation as the property where, under fixed assumptions of validity, a unique explanatory algorithm satisfies the criteria.

Specifically, we ask the following questions: In the where-then-what strategy, (i) is the circuit (the "where") unique? (ii) Is a given circuit's grounding interpretation (the "what") unique? In the what-then-where strategy, (iii) is the causally-aligned algorithm (the "what") unique? (iv) For a given algorithm, is there a unique subspace of the neural network (the "where") that is causally aligned?

We stress-test the identifiability properties of current MI criteria by conducting experiments in a controlled, small-scale setting. Using simple tasks like learning Boolean functions and very small multi-layer perceptrons (MLPs), we search for Boolean circuit explanations – aiming to discover which succession of logic gates is implemented by the MLPs. This setup allows us to exhaustively enumerate incompatible candidate explanations and test them with existing criteria. Our experiments reveal non-identifiability at every stage of the MI process. Specifically, we find that: (i) Multiple circuits can perfectly replicate the model's behavior (with a circuit error of zero), (ii) for a given circuit, multiple valid interpretations exist, (iii) several algorithms can be perfectly causally aligned with the neural computation (IIA of one), and (iv) for a given causally aligned algorithm, multiple subspaces of the neural network can be equally aligned (IIA = 1).

In the discussion, we revisit whether unicity is necessary. We discuss alternative criteria and perspectives that do not require modifying existing criteria, such as requiring explanations only to meet predictive and/or manipulability standards. However, if unicity is considered essential, e.g., to provide a sense of understanding, we also discuss less permissive criteria. Finally, we also refer to the inner interpretability framework (Vilas et al., 2024) that requires an explanation to be validated by multiple complementary criteria. We hope our work contributes constructively to the ongoing effort to develop rigorous definitions for what it means to explain a complex neural network.

## 2 BACKGROUND

### 2.1 MECHANISTIC INTERPRETABILITY

Mechanistic interpretability rests on the key assumptions that a neural network's behavior *can* be explained by a simpler algorithm than the full network, and that a sparse subset of the network executes this algorithm. Previous research has given support to these assumptions: pruning studies (Gale et al., 2019; Ma et al., 2023; Sun et al., 2024) and the lottery ticket hypothesis (Frankle & Carbin, 2019; Liu et al., 2024) suggest that networks are often overparameterized, and only a fraction of neurons and connections are critical to the final performance. Training sub-networks (Yuan et al., 2019) to approximate the full model (Liao & Kyrillidis, 2022), similar to dropout (Srivastava et al., 2014), supports the idea that sub-networks *can* approximate the full network's behavior well.

This search for interpretable circuits is inspired by neuroscience, which has long sought to uncover neural circuits that explain observed behaviors (Yuste, 2008). Once the neural circuit is discovered, researchers focus on interpreting the functional roles of each component in the brain (Yuste, 2008). Research in computer vision has already shown that some nodes within neural networks compute

interpretable features (Olah et al., 2017). Connections between such features, also called *circuits*, can be compact explanations of model behavior (Olah et al., 2020; Carter et al., 2019; Dreyer et al., 2024). Finally, recent work has applied mechanistic interpretability to LLMs (Elhage et al., 2021), especially in transformer models (Templeton et al., 2024; Bricken et al., 2023; Vilas et al., 2023). For example, Wang et al. (2022) identified a circuit responsible for Indirect Object Identification (IOI) in transformers, highlighting the potential for mechanistic explanations of complex LLM behaviors.

## 2.2 Definitions

A satisfactory mechanistic explanation of a model's behavior consists of two components: the *what*, a high-level algorithm that closely approximates the model's behavior and tracks its internal computation, and the *where*, specifying how and where this algorithm is embedded in the model's low-level neural computation.

We refer to the combination of an explanatory algorithm and the mapping between the high- and low-level states as a *computational abstraction*. This is an abstraction as it simplifies the neural network's computation, focusing on a subset of the computational graph and abstracting neural activations into simpler, high-level features. For example, consider the mechanistic explanation of how a vision algorithm recognizes rectangles. We might identify a computational abstraction where certain modules perform edge detection, others detect right angles, and a final component applies an AND logic gate to confirm the presence of four right angles. This abstraction specifies the algorithm and how and where low-level neural activations correspond to high-level features of the algorithm. In this work, we interchange the terms *explanation* and computational abstraction.

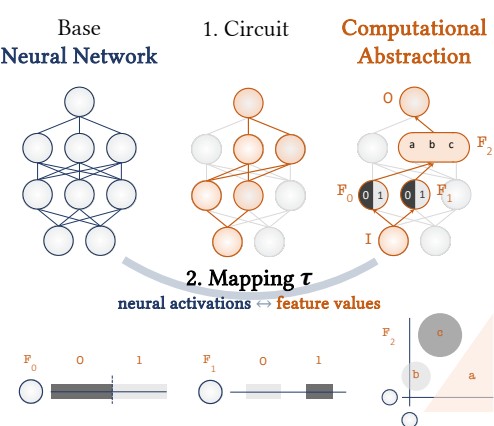

Figure 1: Illustration of the computational abstraction components within a neural network. The **circuit** represents a subgraph, and the mapping specifies the high-level **features** computed by the circuit, detailing how their values arise from low-level neural activations. Together, these form the computational abstraction (explanation of the neural network). Here, feature $F_2$ has three possible values and is defined within the 2D activation space of two neurons. Features $F_0$ and $F_1$ are binary variables, each assigned to a single neuron. $F_0$ covers the entire activation space and $F_1$ only maps specific intervals, leaving some activations unassigned.

Formally, we define a computational abstraction $A$ as a tuple $(S, \tau)$, where $S$ is the *circuit*, the subset of the neural network's computational graph responsible for the behavior of interest, and $\tau$ is the mapping between the states of the circuit and the states of the variables of the algorithm, which specifies how to interpret the computational function of the circuit's components. We now proceed to define the circuit and mapping formally.

**Definition 1** (Circuit). *Let $G = (V, E)$ represent the computational graph of a neural network, where $V$ is the set of nodes (neurons) and $E \subseteq V \times V$ is the set of edges (connections between neurons). A circuit $S = (V_S, E_S)$ is a subgraph of $G$ that contains at least one path from a subset of input nodes to a subset of output nodes.*

**Definition 2** (Mapping ($\tau$)). *A mapping between low-level values taken by neurons and high-level values taken by the variables of the explanatory algorithm consists of a set of $K$ surjective maps, one for each high-level variable. Each associates the neural network activations with the values of the corresponding high-level variable. For a group of neurons $V_j$ in the neural network, mapped to a high-level variable $A_j$ with possible values $\{f_0, \ldots, f_m\}$, the mapping $\tau_j : \mathbb{R}^{|V_j|} \to \{f_0, \ldots, f_m\}$ assigns a vector of activations to one of the possible values of $A_j$. Each mapping should be surjective $\left(\forall f_i, \exists h \in \mathbb{R}^{|V_j|} : \tau_j(h) = f_i\right)$ and with a non-empty pre-image $\left(\forall f_i, \tau_j^{-1}(f_i) \neq \emptyset\right)$*

*These conditions ensure that all high-level values can be realized by some set of low-level activations.*

In practice, we are interested in mappings that satisfy a *consistency* requirement. Intuitively, consistency means that if we first perform part of the computation using the neural network and then apply the mapping to get the state of a high-level variable, the outcome should be identical to applying the mapping and then performing the computation of the high-level algorithm. The computations in the neural network and the high-level algorithm should align consistently according to the mapping.

**Definition 3** (Consistent Mapping). *Let $\tau$ be a mapping between groups of low-level neurons $\{V_j\}$ and their corresponding high-level variables $\{A_j\}$. The mapping $\tau$ is said to be consistent if for any high-level variable $A_j$, with parents $PA_j$, the following diagram commutes:*

$$
\begin{array}{ccc}
PA_j & \xrightarrow{\;Alg.\;} & A_j \\[2pt]
\Big\uparrow{\scriptstyle \tau_{PA_i}} & & \Big\uparrow{\scriptstyle \tau_j} \\[2pt]
\mathbb{R}^{|V_{PA_j}|} & \xrightarrow{\;NN\;} & \mathbb{R}^{|V_j|}
\end{array}
$$

*Here: $\tau_{PA_j}$ represents the application of $\tau$ to each variable in $PA_j$; $NN$ refers to the computation between the low-level neural network states; and $Alg.$ refers to the computation between high-level variables governed by the explanatory algorithm.*

Previous works have explored various types of high-level features and representational abstractions, including mappings based on "directions in activation space" or specific points within activation subspaces (Olah et al., 2020; 2018; Bereska & Gavves, 2024). This work focuses on explanatory algorithms represented as Boolean circuits, where high-level features are binary (0 or 1). The mappings specify which activations correspond to 0 and 1. Boolean circuits are computationally universal and thus sufficient to demonstrate identifiability issues in existing MI criteria.

## 2.3 Approaches to Circuit Discovery

We identify and describe two strategies for reverse-engineering neural networks: the *where-then-what* and *what-then-where* approaches.

### Where-then-what

Methods from this strategy first aim to identify a circuit that replicates the behavior of the full model well. Once a circuit is found, the next step is to interpret its components to uncover the high-level algorithm being implemented (Dunefsky et al., 2024; Davies & Khakzar, 2024). The evaluation criteria for circuits is how well they replicate the full model's behavior for the input of interest.

**Definition 4** (Circuit Error). *Let $S$ be the function computed by a circuit and $g$ the function computed by the model on which the circuit is defined. For the input set $\mathbf{x}$, the error of the circuit $S$ is: $1 - \frac{1}{|\mathbf{x}|} \sum_{x \in \mathbf{x}} \mathbb{1}[S(x) = g(x)]$*

In the case of perturbed inputs, it can also be defined via the KL divergence between the logits of the circuit and the model (Conmy et al., 2023).

In practice, circuit search relies on causal mediation analysis, which seeks to isolate the subset of the network that carries the information from the inputs to the output. Since it is computationally intractable to enumerate all possible circuits in complex models (Adolfi et al., 2024b), existing methods focus on computing mediation formulas for individual components to decide their inclusion in the circuit (Vig et al., 2020; Meng et al., 2022; Monea et al., 2024; Kramár et al., 2024; Conmy et al., 2023; Geva et al., 2023; Syed et al., 2023).

A combination of data analysis and human input is typically used to interpret candidate circuits. For example, activation maximization identifies inputs that maximally activate a component, which helps clarify its function (Zhou et al., 2016; Zeiler & Fergus, 2014; Simonyan et al., 2014). This technique has been extended to modern LLMs (Peyrard et al., 2021; Jawahar et al., 2019; Dai et al., 2022). However, polysemantic neurons, which encode multiple concepts simultaneously (Templeton et al., 2024; Bricken et al., 2023), complicate the interpretation of LLMs components. For a broader overview of these challenges, we refer readers to the following surveys: Sajjad et al. (2022); Khakzar et al. (2021). In this work, we use the concept of consistent mapping as the objective evaluation of the quality of an interpretation.

WHAT-THEN-WHERE

Methods from this strategy first hypothesize a candidate high-level algorithm and then search for mappings between the states of this algorithm and subspaces of the neural activations. The goal is to identify mappings where the high-level and low-level states are causally aligned, meaning they respond similarly under interventions.

Given a candidate high-level algorithm $A$, neural activations $H$, and a mapping $\tau$ defined between them, counterfactual interventions are performed on the inner variables of $A$, and corresponding interventions are applied to $H$ via $\tau$. *Intervention interchange accuracy* (IIA) (Geiger et al., 2022b) is then defined for each high-level variable and measures the similarity of outputs in $A$ and $H$ after intervening (metric for causal alignment). We give in Appendix A a complete, formal definition.

A perfect IIA score (1) for all variables indicates that all possible interventions produce the same effect in low-level and high-level models. In practice, exhaustive enumeration is often impractical, and IIA is approximated using randomly sampled inputs (Geiger et al., 2022b). Similarly to the mapping consistency defined above, perfect causal alignment requires diagram commutation between low- and high-level models under interventions.

Searching for causal alignment between high-level models and neural activations can be computationally expensive, as it often requires testing many potential mappings. The Distributed Alignment Search method (Geiger et al., 2024) addresses this challenge by employing gradient descent to search for alignments efficiently. This approach also allows for distributed representations, where multiple neurons represent a single high-level variable. Indeed, an underlying assumption of IIA is that the neural activations corresponding to distinct high-level variables are disjoint: $\forall x, y \in V, \tau^{-1}(x) \cap \tau^{-1}(y) = \emptyset$, which may not occur in real-world examples (Olah et al., 2020).

Currently, no systematic method exists for choosing which candidate algorithms to test. Previous work (Wu et al., 2023) has manually proposed a few candidates, but the vast space of possible algorithms makes this an open challenge.

## 2.4 EXPLANATION "IDENTIFIABILITY"

The assumption of explanatory unicity – the idea that there exists a single, unique explanation for a given phenomenon – is not only implicit in the practice of mechanistic interpretability (see relevant citations in Appendix C) but also rooted in human cognitive and psychological tendencies (Trout, 2007; Waskan, 2024; Gopnik, 2000). Humans demonstrate a cognitive preference for coherent explanations that integrate disparate observations into a unified narrative (e.g., Friedman, 1974; Kitcher, 1962; 1981; Schurz, 1999; Kveraga et al., 2007). This preference aligns with the psychological need for cognitive closure, defined as the desire for a definitive conclusion (Kruglanski, 1989). Multiple incompatible explanations disrupt coherence, leading to ambiguity and a sense of unresolved understanding.

Conversely, in the philosophy of science, explanatory pluralism acknowledges that the world is too complex to be fully described by a single comprehensive explanation (Kellert et al., 2006; Potochnik, 2017). Multiple explanations often coexist without conflict because they address different explanatory goals (e.g., explaining distinct behaviors) or employ different simplification strategies (e.g., differing levels of abstraction Marr & Poggio, 1976). However, in this work, we deliberately search for conflicting explanations by fixing both the explanatory goal and the simplification strategies, as the ones defined by MI criteria.

**Identifiability and incompatible explanations.**

As mentioned in Section 1, we borrow the term of identifiability from the field of statistics (Rothenberg, 1971), defining *identifiability of explanation* as the property where a unique explanation is valid under fixed standards of validity. An MI strategy is not identifiable if its standards of validity do not discriminate between two incompatible explanations.

We define two explanations as **incompatible** or **conflicting** if they share the same explanatory goal and simplification strategy, but posit different computational abstractions. In our context, the explanatory goal is fixed: explaining the specific input-output behavior of a trained MLP. The simplification strategy is also fixed, corresponding to one of two predefined strategies to find computational abstractions: the what-then-where or the where-then-what defined above.

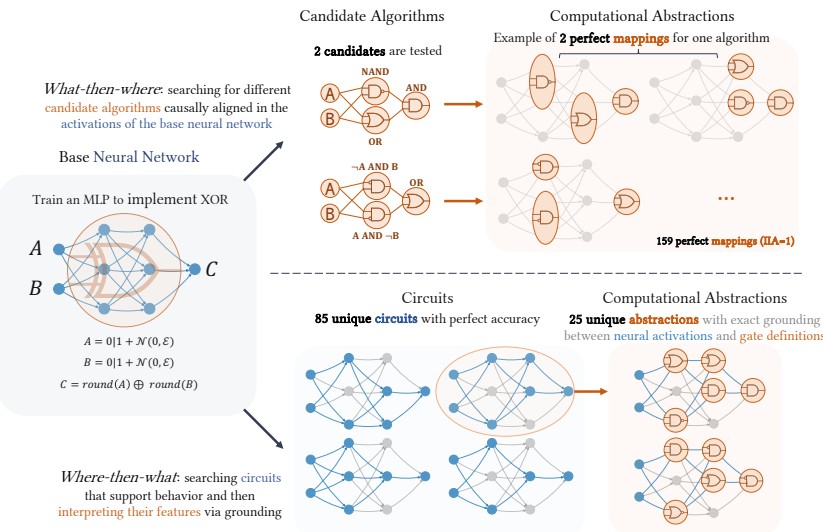

Figure 2: **Illustration of identifiability problems using the XOR example.** We train a small MLP with two hidden layers of size 3 to compute the XOR function perfectly. The figure shows the outcome of stress-testing the two reverse-engineering strategies: Top: For the *what-then-where* strategy, we enumerate all subsets of neurons searching for subsets causally aligned with intermediate variables of candidate algorithms, with alignment measured by IIA. Even testing only two candidate algorithms, we find perfect implementations of both in the model. Multiple mappings (localizations) for each algorithm were identified, showing that neither the algorithm (*what*) nor its location in the network (*where*) is unique. Bottom: For the *where-then-what* strategy, we enumerate circuits (sub-networks) and test whether each computes the XOR independently. For each circuit, we search for possible feature interpretations of the selected neurons, identifying intermediate logic gates whose values can be mapped consistently with the neurons' activations. Consistency is defined as in 3. We find many different perfect circuits (the *where* is not unique) and for any given circuit, we find multiple valid interpretations (the *what* is not unique).

Incompatibility of two explanations can occur if (1) the two explanations posit different algorithms for the same behavior, or (2) the same algorithm is embedded in different subspaces of the neural network. Both scenarios entail different internal representations and causal pathways linking inputs to outputs. In Figure 2, we report examples of incompatible explanations in trained MLPs.

Our experiments show that even the strict causal criteria of MI allow many incompatible computational abstractions. In the discussion (Section 5), we revisit whether this expectation of unicity is necessary or even achievable.

## 3 Illustrating Potential Identifiability Issues

This section highlights identifiability counter-examples for a small MLP trained to compute the XOR function. It is well-known that an MLP requires at least two layers to compute the XOR function. Once the network can do so, the interpretability exercise becomes: how is the XOR implemented? For a mechanistic explanation, the answer must have two components: *what* algorithm is being used, such as which combination of logic gates transforms the inputs into the XOR truth table, and *where* these intermediate logic gates are located within the neural network's computation—i.e., *where* the algorithm is executed within the MLP.

To stress-test the two main MI strategies (*where-then-what* and *what-then-where*), we chose an MLP small enough to allow exhaustive enumeration of all circuits and extensive search over mappings. The MLP is trained on binary inputs with a single logit output to produce the XOR behavior. The inputs are 0 or 1 and can have a randomly sampled Gaussian noise of a fixed standard deviation.

Our methods to test the different criteria defined in the previous section are as follows:

**Circuit search:** We enumerate all possible circuits, and then execute the validation data of the XOR on each circuit as if it were a standalone neural network, effectively removing from the computation each node and edge that is not part of the circuit. If a circuit achieves perfect accuracy (zero circuit error), we label it a *perfect circuit*, as it exactly replicates the model's behavior. This search tests the identifiability property of the circuit error criteria.

**Interpretation search:** For each perfect circuit, we attempt to interpret the activations of the included neurons based on XOR validation data. As the scope of interpretations is limited to logic gates, we search, for each neuron, a logic gate whose values are consistent with that neuron's activation. The method proceeds recursively, layer by layer, based on a given neuron's relationship with its parents in the circuit. The parents already have an interpretation (mapping their activations to 0 or 1). We enumerate all possible inputs from the parents and examine how they are mapped into the neuron's activation by the model. We then list all possible ways to separate these inputs and label the resulting logic gate. If we find no valid interpretation for a given neuron (e.g., all inputs overlap in the output activation and no separation is possible), we end this candidate interpretation of the circuit. If we find multiple, we expand the tree of possible candidate interpretations for the circuit. To avoid trivial over-counting, we ignore value relabeling (e.g., swapping 0 and 1) and, by convention, assign 1 to the larger intervals and 0 to the smaller ones. The outcome is a computational abstraction, a Boolean circuit computing the XOR function whose internal logic gates are mapped to some neural network components. Note that this method undercounts possible interpretations because it does not consider cases where the high-level logic gates are mapped on multiple low-level neurons. This search tests the identifiability property of the mapping consistency criteria.

**Algorithm search:** We enumerate all algorithms implementing the XOR gate by considering Boolean formula parse trees. We assume that each neuron's activations encode either the identity, AND, or OR gate, a network of depth $d$ can implement only formulas with a parse tree of depth $\leq d$. We therefore recursively enumerate all unique formulas of depth $d$ or less using AND, OR, and negation. For $d = 3$, this results in 56 XOR-equivalent algorithms.

**Mapping search:** For a given candidate algorithm with specified intermediate logic gates, we explore all possible neuron subsets and mappings between these subsets and the algorithm's intermediate gates. We then measure the causal alignment of the mapping using IIA. If a mapping achieves perfect IIA, we call it a *perfect* mapping. If there is no other mapping with larger images (set inclusion-wise), we also call this mapping *minimal*. In the example described in this section, we manually test two candidate algorithms, while the next section enumerates algorithms that implement the target function, excluding trivial variations (e.g., negating gates). This search tests the identifiability property of the IIA criteria.

We depict in Figure 2 counter-examples for each criterion in one small MLP. In this example, for the *what-then-where* strategy, we only test two candidate algorithms but find 159 perfect minimal mappings within the neural network activations, with perfect mappings for both algorithms. Therefore, the algorithm is not unique and, for a given algorithm, its localization is not unique. For the *where-then-what* strategy, we find 85 unique circuits with perfect accuracy, with an average of 535.8 logic gate interpretations (consistent mappings) per circuit. Therefore, the localization is not unique, and for a given circuit, the interpreted algorithm is not unique. Overall, in this example, we obtain 159 + 45,543 computational abstractions, most of which are incompatible. This is a serious identifiability problem as there is no clear and consensual criterion to decide among all these explanations.

# 4 EXPERIMENTS

## 4.1 QUANTITATIVE ANALYSIS

We now repeat the experiment used for the XOR example with different seeds, while varying the architecture size and the complexity of the global behavior.

The basic setup is consistent across all experiments. We choose $n$ 2-input logic gates $L_1, \ldots, L_n$, generate a multilayer perceptron (MLP) $N$ with layer sizes $(2, k, k, n)$, and train $N$ to implement the gates $L_1, \ldots L_n$. Similarly to the previous section, training is performed on binary samples with added Gaussian noise and continues until the network' mean squared loss is lower than $n \times 10^{-3}$.

We then quantify the identifiability issues again. For the circuit-first search (*where-then-what* strategy), we count perfect circuits for $L$ in $N$ and valid interpretations for each circuit. For the algorithm-first search (*what-then-where* strategy), we count perfectly aligned algorithms for $L$ in $N$ and perfect minimal mappings for each algorithm.

In the circuit-first search, we only search for circuits containing two inputs and one output for each target gate. Furthermore, due to the combinatorial explosion in circuit enumeration, we only test circuits with a sparsity greater than 0.3, where sparsity is measured as the fraction of components excluded from the circuit. This number is chosen as the smallest sparsity that remains manageable for the size of MLPs that we consider. As a result, the reported number of circuits should be considered a lower bound. Furthermore, the number of potential interpretations for a single circuit grows exponentially with its size. Since we count interpretations for only the sparser circuits, the reported number of interpretations is also significantly lower than an exhaustive search would yield.

### 4.1.1 ARCHITECTURE SIZE

Increasing the neural network's size may impact the number of computational abstractions found in networks. Although a larger architecture may create more computational abstractions due to the increased search space, it could also lead to greater overparameterization, meaning a smaller subset of the network may suffice to implement the target gate. This, in turn, could reduce the number of valid abstractions if most of the network is inactive during inference.

In the left side of Figure 3, we report the total number of explanations found when the architecture size ranges from $k = 2$ to $k = 5$. We exclude from this figure the networks for which no valid mapping or interpretation was found, which we give in Appendix D.2 along with additional plots.

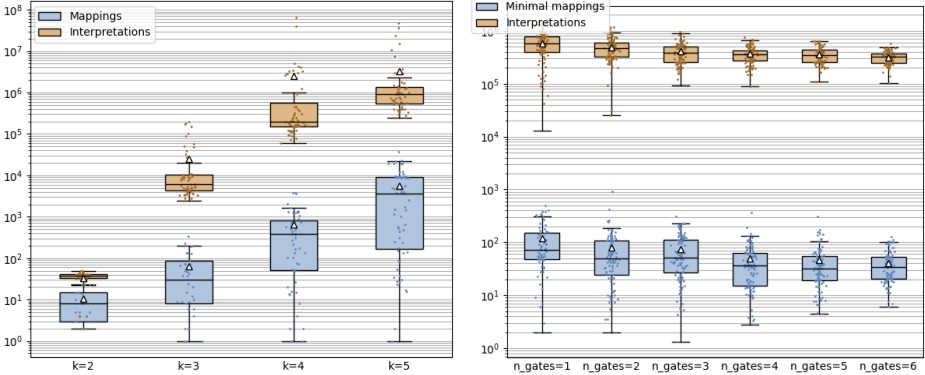

Figure 3: Number of computational abstractions found in the circuit-first approach (circuit interpretations) and the algorithm-first approach (perfect minimal mappings), as a function of architecture size $k$ (left) or of the number $n$ of gates the model is trained on (right, averaged over all target gates). One point per neural network.

In both cases, we observe that the number of computational abstractions found significantly increases with network size, with median values growing from 38 to 910,000 in the circuit-first method and from 8 to 3,700 in the algorithm-first approach. Less than 2% of the trained networks contain exactly one valid minimal mapping, and no network contains exactly one circuit interpretation.

### 4.1.2 MULTIPLE TASKS

We also investigate the effect of global behavior complexity. The model is trained to implement a single logic gate in the basic setup. What happens when the network is trained in a multi-task setting? As the number of target tasks increases, we expect the network to use its activations more efficiently, possibly relying on a smaller subset of its structure for each task. To explore this, we fix $k = 3$ and vary $n$ from 1 to 6, sampling $n$ logic gates (without replacement) from the same list as above, extended to include the negation of the gates (NOR, NAND, NIMP, and XNOR). The neural network $N$ is then trained to implement these gates in parallel, using two shared input neurons and $n$ output neurons (one per gate). We repeat this procedure using different random seeds.

The right side of Figure 3 contains the total number of computational abstractions obtained when training each neural network on a different number of logic gates in parallel, ranging from 1 to 6. More detailed plots are available in Appendix D.3.

In both approaches, the number of interpretations significantly decreases with the number of training tasks ($p = 0.05$), up to 4 tasks. Past that point, the variation is no longer statistically significant.

## 4.2 TRAINING DYNAMICS

A possible explanation for the high number of computational abstractions we find in trained networks is that our networks do not perfectly implement the target logic gates, since training is stopped when a low but non-zero loss value is reached. We explored this effect by varying the loss cutoff in the basic setup. The results are given in Appendix D.4. More generally, the influence of the training distribution on the number of valid computational abstractions is investigated in Appendix D.6. In addition, we find that adding noise to binary samples during training has no significant effect on the results obtained in the algorithm-first method, but decreases the number of circuits while increasing the overall number of interpretations found in the circuit-first method (results are described in Appendix D.5). Training dynamics and generalization abilities of a model may therefore reduce the number of available abstractions, but this effect alone is unlikely to mitigate the issue entirely.

## 4.3 TOWARDS LARGER MODELS

While enumerating circuits or mappings is infeasible in large networks, it is still possible to find counterexamples in which multiple circuits exist. For example, we trained a larger MLP on a subset of the MNIST dataset (Deng, 2012), filtered to contain only the digits 0 and 1. We obtained a regression model with layer sizes (784, 128, 128, 3, 3, 3, 1). After training, we extracted the last layers of the model to form two sub-networks: one of size (784, 128, 128, 3) and one of size (3, 3, 3, 1). We fed the training samples through the larger sub-network, generating a new dataset comprised of partial computations of the overall model.

Applying the circuit search method to the smaller sub-network using this new dataset yielded 3,209 valid circuits. While we cannot enumerate circuits in the first half of the network, any such valid circuit can include one of the circuits of the second half as its continuation. Two situations may arise: If the first half of the MLP does not contain any valid circuits, then no valid circuit exists for the full network; if valid circuits exist in the first half of the MLP, then a minimum of 3,209 valid circuits exist in the full network. This shows, at least in the case of circuits, that the problem does not seem to disappear with significantly larger scale and more complex data distributions.

## 5 WHAT DOES IT MEAN FOR INTERPRETABILITY?

Our findings challenge the strong intuition that a unique mechanistic explanation exists for a given behavior under fixed explanatory goals and validity criteria. Even when employing the strict causal requirements of MI, we find that many incompatible explanations can coexist. While predictive of behavior and causally aligned with the neural network's states, these explanations differ in the computational algorithms they postulate or how they are embedded in the network's subspaces. We now discuss several ways to move forward from this striking observation.

## 5.1 DOES LACK OF UNICITY MATTER?

Whether multiple "valid" explanations pose a real problem is worth considering. From a pragmatic stance, one could argue that unicity is not essential if the explanations meet functional goals such as predictivity, controllability, or utility in decision-making (Van Fraassen, 1988; Achinstein, 1984). This perspective emphasizes crafting practical criteria to evaluate explanations based on their utility, rather than their ontological closeness to the *truth*. Stating explicitly the pragmatic goals of an explanation can also clarify what is expected of an explanation (Woodward & Ross, 2021). For example, in the recent debate about *interpretability illusion*, Makelov et al. (2023) mention problems about interventions that can potentially activate *dormant pathways* leading the resulting explanation to *misrepresent* the mechanisms at play. In their response, one of the arguments advanced by Wu et al. (2024) is to point out that the explanation produced by their method (DAS), still meets the

pragmatic goals of predictivity and manipulability, ensuring its usefulness. The debate is resolved by clarifying the epistemic goals of the explanation.

## 5.2 IF YES, HOW CAN WE AIM TO RESOLVE IT?

If we decide that identifiability of explanations is important, our work demonstrates that current MI criteria are insufficient to guarantee it. One potential approach to resolving this issue involves introducing additional heuristics, such as prioritizing the sparsest circuits. However, Occam's razor alone is unlikely to solve the problem. Should we dismiss an entirely different candidate explanation simply because it involves one additional node than another? In our experiments, simplicity or sparsity cannot single out one explanation.

To address these challenges, we believe that ideas from causal abstraction (Beckers & Halpern, 2019; Beckers et al., 2020; Rubenstein et al., 2017) can be helpful (Geiger et al., 2022a). Although IIA is directly inspired by causal abstraction, it does not fully implement it in its current form. Unlike current MI frameworks, causal abstraction requires that all lower-level model states are accounted for in higher-level representations. Furthermore, if components are excluded from an explanation, their absence must be justified causally. This intuition has been formalized recently through the concept of faithfulness, which evaluates how well a circuit replicates the model's behavior and the (lack of) impact of excluded elements (Hanna et al., 2024).

Alternatively, one can look at broader approaches and not focus on searching for explanations that optimize a single criterion. Interestingly, Vilas et al. (2024) propose an *inner interpretability framework* based on lessons from cognitive neuroscience. In this framework, the authors emphasize the importance of building multi-level mechanistic explanations and stress-testing these explanations with proper hypotheses testing. An explanation validated by many criteria and which exhibits different properties (e.g., invariances) becomes more trustworthy. This framework provides a promising path to establish MI as a natural science akin to neuroscience or biology.

## 5.3 IS IDENTIFIABILITY EVEN ACHIEVABLE?

In some domains of science, competing theories coexist despite being ontologically incompatible. For instance, the Lagrangian and Hamiltonian formulations of classical mechanics posit different underlying entities but yield identical experimental predictions. Such scenarios are examples of *contrastive underdetermination*, a philosophical debate about whether empirical evidence *can* or *should* uniquely determine scientific explanations (Stanford, 2023). The intrinsic computational hardness of interpretability queries (Adolfi et al., 2024b;a) suggests that MI may have fundamental limits, leaving it possibly underdetermined. More broadly, the search for MI explanations can be seen as a type of causal representation learning, where identifiability may depend on strong constraints that might not be realistic Morioka & Hyvärinen (2024).

## 5.4 FROM TOY MODELS TO REAL MODELS

Our experiments focus on toy MLPs trained on toy tasks, which differ drastically from large language models (LLMs) trained on vast, complex datasets using Transformer architectures. This raises the possibility that the issues in small models may not apply to larger, more sophisticated models. However, if this is true, why the problems would disappear at larger scales must be demonstrated. Understanding why current criteria function well in some regimes but not in others would also lead to refined criteria and definitions.

## 5.5 CONCLUSION

Our results should encourage the community to reflect on the role of unicity when searching for and communicating about mechanistic explanations found in neural networks. We believe that exploring stricter criteria based on causal abstraction, explicitly formulating pragmatic goals of explanations and embracing broader frameworks such as the inner interpretability one are all promising directions.

The code and parameters used to conduct this paper's experiments can be found on GitHub.

ACKNOWLEDGEMENTS

This work was conducted within French research unit UMR 5217 and was partially supported by CNRS (grant ANR-22-CPJ2-0036-01) and by MIAI@Grenoble-Alpes (grant ANR-19-P3IA-0003). It was granted access to the HPC resources of IDRIS under the allocation 2025-AD011014834 made by GENCI.

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

## A    FORMAL DEFINITION OF IIA

The definitions in this section are adapted from Geiger et al. (2022b). We begin by setting notation conventions.

Let $N$ be a neural network, and $A$ be a high-level algorithm.
Let $\tau$ be a mapping between low-level neuron groups $\{V_j\}$ of $N$ and the values of their corresponding high-level variables $\{A_j\}$ in $A$.
Let $V_{\text{in}}$ (resp. $V_{\text{out}}$) be the neuron groups corresponding to the inputs (resp. outputs) of $N$.
Let $A_{\text{in}}$ (resp. $A_{\text{out}}$) be the variables in $A$ with no parents (resp. no children).

**Notation** (Value reading). *Let $v_{in} \in \mathbb{R}^{|V_{in}|}$ be some possible input values of the network $N$. We note $N[v_{in}, V_j]$ the values of the activations of $V_j$ that are obtained when setting the values of $V_{in}$ to $v_{in}$ and running the computation graph of $N$.*
*Similarly, let $a_{in} \in \mathbb{R}^{|A_{in}|}$ be some possible values of the variables of $A$ with no parents. We note $A[a_{in}, A_j]$ the values of the variable $A_j$ that are obtained when setting the values of $A_{in}$ to $a_{in}$ and running the algorithm $A$.*

**Notation** (Intervention). *Let $v_j \in \mathbb{R}^{|V_j|}$ be some possible activations of the variable $V_j$ in the network $N$. We note $N_{V_j \leftarrow v_j}$ a copy of $N$, in which the activations of $V_j$ are forcibly set to the value $v_j$ during the computation.*
*Similarly, let $a_j \in \mathbb{R}^{|a_j|}$ be a possible value of the variable $A_j$ in the algorithm $A$. We note $A_{A_j \leftarrow a_j}$ a copy of $A$, in which the value of $A_j$ is forcibly set to the value $a_j$ when the algorithm is run.*

This notion is aligned with the *do*-operator (Pearl, 2009).

**Definition 5** (Intervention interchange). *Let $base_{l,in}, source_{l,in} \in \mathbb{R}^{|V_j|}$ be some possible input values of the network $N$. We call low-level intervention interchange the quantity:*

$$II_{low}(N, base_{l,in}, source_{l,in}, V_k) = (N_{V_k \leftarrow N[source_{l,in}, V_k]})[base_{l,in}, V_{out}]$$

*Similarly, let $base_{h,in}, source_{h,in} \in \mathbb{R}^{|A_j|}$ be some possible values of the variables in $A$ with no parent. We call high-level intervention interchange the quantity:*

$$II_{high}(N, base_{h,in}, source_{h,in}, A_k) = (A_{A_k \leftarrow N[source_{h,in}, A_k]})[base_{h,in}, A_{out}]$$

This corresponds to the notion of counterfactual intervention: After running the algorithm (or network) on a set of inputs (source) and recording the value of a given variable, we execute the algorithm again on a different set of inputs (base) but restore the value of the variable from the first run during the computation. The system is now in a counterfactual state, and we measure its new output.

**Definition 6** (IIA). *Let $Val(A_j)$ be the set of possible values of $A_j$, and $Val(A_{in}) = \prod_{V \in V_{in}} Val(V)$ be the set of possible combinations of values of the variables in $A$ with no parents. Let $A_k$ be a high-level variable of $A$.*

*The intervention interchange accuracy of the mapping $\tau$ for the variable $A_k$ is the quantity:*

$$IIA(N, A, A_k, \tau) = \frac{1}{|Val(A_{in})|^2} \sum_{b,s \in Val(A_{in})} \mathbb{1}\left[II_{high}(A, b, s, A_k) = II_{low}(N, b, s, V_k)\right]$$

## B    OUTPUT EXAMPLES

This section contains additional examples of computational abstractions found by both strategies. Specifically, we report abstractions found in a single neural network trained on the XOR gate with $k = 3$ and $n = 1$ with a loss cutoff of $10^{-3}$.

## B.1 CIRCUIT-FIRST APPROACH

An exhaustive pass of the circuit-first approach (with no minimal sparsity threshold) yielded 59 circuits (*where*). We depict in Figure 4 the 12 most sparse circuits found.

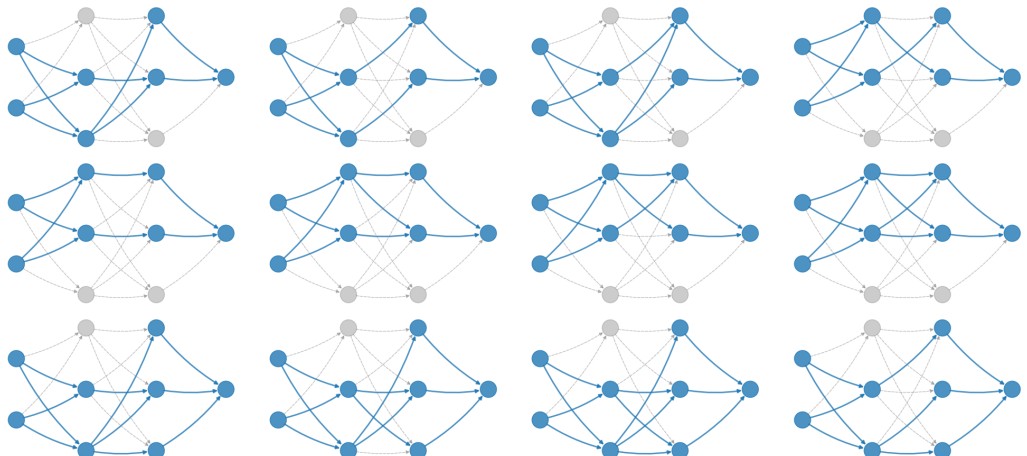

Figure 4: The 12 most sparse circuits found in the example network.

When searching for interpretations (*what*) in all 59 circuits, we find 114,230 interpretations. We then focus on circuit 8 (second row, last column in Figure 4, chosen for illustration purposes as it yields 24 valid interpretations, which we fully list in Table 1. Each of these interpretations leads to a different explanation of the network; for example, interpretation 1 corresponds to the formula $\neg(\neg(A \wedge B) \to \neg(A \vee B))$, while interpretation 2 corresponds to $\neg((\neg(A \wedge B) \vee \neg(A \vee B)) \to (\neg(A \wedge B) \to \neg(A \vee B)))$.

## B.2 ALGORITHM-FIRST APPROACH

In the algorithm-first approach, exhaustive enumeration yields 56 possible logic formulas for the XOR gate with a depth of 3 (excluding commutative-invariant formulas). Four of these formulas produce valid mappings for the neural network. Those mappings are listed in Table 2.

## C IDENTIFIABILITY IN CIRCUIT LITERATURE

Identifiability is, to the best of our knowledge, never stated as an explicit assumption in existing works about circuits. In this section, we list examples from the literature that indicate that it is nonetheless typically taken for granted:

- Cammarata et al. (2021): "the curve circuit" (multiple occurrences)

- Wang et al. (2022): "discover the circuit", "discovering the circuit", "uncover the circuit"

- Kramár et al. (2024): "we investigate the circuit underlying multiple-choice question-answering"

- Conmy et al. (2023): "Choosing a clearly defined behavior means that the circuit will be easier to interpret than a mix of circuits corresponding to a vague behavior", "ACDC [...] fully recovers the circuit of toy model".

- Hanna et al. (2024): "We next search for the circuit responsible for computing this task"

- Marks et al. (2024): "The circuit for agreement across a prepositional phrase (Figure 12)"

| # | Neuron (1, 0) Gate | Sep. | Neuron (1, 1) Gate | Sep. | Neuron (2, 0) Gate | Sep. | Neuron (2, 1) Gate | Sep. | Neuron (3, 0) Gate | Sep. |
|---|---|---|---|---|---|---|---|---|---|---|
| 1 | -0.003 | NAND | 0.693 | NOR | 0.151 | IMP | 0.777 | OR | 0.5 | NOT A |
| 2 | -0.003 | NAND | 0.693 | NOR | 0.151 | IMP | 0.777 | OR | 0.5 | RNIMP |
| 3 | -0.003 | NAND | 0.693 | NOR | 0.151 | IMP | 0.777 | A | 0.5 | NOT A |
| 4 | -0.003 | NAND | 0.693 | NOR | 0.151 | IMP | 0.777 | A | 0.5 | RNIMP |
| 5 | -0.003 | NAND | 0.693 | NOR | 0.151 | IMP | 0.987 | NIMP | 0.5 | IMP |
| 6 | -0.003 | NAND | 0.693 | NOR | 0.151 | IMP | 0.987 | NIMP | 0.5 | NOT A |
| 7 | -0.003 | NAND | 0.693 | NOR | 0.151 | IMP | 0.987 | NIMP | 0.5 | B |
| 8 | -0.003 | NAND | 0.693 | NOR | 0.151 | IMP | 0.987 | NIMP | 0.5 | RNIMP |
| 9 | -0.003 | NAND | 0.693 | NOR | 0.397 | NOT A | 0.987 | NIMP | 0.5 | B |
| 10 | -0.003 | NAND | 0.693 | NOR | 0.397 | NOT A | 0.987 | NIMP | 0.5 | RNIMP |
| 11 | -0.003 | NAND | 0.693 | NOR | 0.397 | NOR | 0.987 | NIMP | 0.5 | B |
| 12 | -0.003 | NAND | 0.693 | NOR | 0.397 | NOR | 0.987 | NIMP | 0.5 | RNIMP |
| 13 | 0.321 | NOR | 0.230 | NAND | 0.151 | RIMP | 0.777 | OR | 0.5 | NOT A |
| 14 | 0.321 | NOR | 0.230 | NAND | 0.151 | RIMP | 0.777 | OR | 0.5 | RNIMP |
| 15 | 0.321 | NOR | 0.230 | NAND | 0.151 | RIMP | 0.777 | B | 0.5 | NOT A |
| 16 | 0.321 | NOR | 0.230 | NAND | 0.151 | RIMP | 0.777 | B | 0.5 | RNIMP |
| 17 | 0.321 | NOR | 0.230 | NAND | 0.151 | RIMP | 0.987 | RNIMP | 0.5 | IMP |
| 18 | 0.321 | NOR | 0.230 | NAND | 0.151 | RIMP | 0.987 | RNIMP | 0.5 | NOT A |
| 19 | 0.321 | NOR | 0.230 | NAND | 0.151 | RIMP | 0.987 | RNIMP | 0.5 | B |
| 20 | 0.321 | NOR | 0.230 | NAND | 0.151 | RIMP | 0.987 | RNIMP | 0.5 | RNIMP |
| 21 | 0.321 | NOR | 0.230 | NAND | 0.397 | NOT B | 0.987 | RNIMP | 0.5 | B |
| 22 | 0.321 | NOR | 0.230 | NAND | 0.397 | NOT B | 0.987 | RNIMP | 0.5 | RNIMP |
| 23 | 0.321 | NOR | 0.230 | NAND | 0.397 | NOR | 0.987 | RNIMP | 0.5 | B |
| 24 | 0.321 | NOR | 0.230 | NAND | 0.397 | NOR | 0.987 | RNIMP | 0.5 | RNIMP |

Table 1: The list of interpretations found for circuit 8 of 59 in the example network. For each interpretation, the four intermediate neurons and the output one are assigned a logic gate and a separation boundary. Each neuron is represented by its layer and position in the layer (indexed from 0), as read from left to right and from top to bottom in Figure 4. IMP refers to the implication gate (A implies B), NIMP to its negation, and RIMP and RNIMP refer to the reversed implication gate (B implies A) and its negation.

**Formula 1:** $\neg(A \wedge B) \wedge (A \vee B)$

| Mapping | $A \vee B$ | $\neg(A \wedge B)$ |
|---|---|---|
| 1 | Neuron (1, 2) | Neuron (1, 1) |
| 2 | Neuron (1, 0) | Neuron (1, 1) |

**Formula 2:** $\neg((A \wedge B) \vee \neg(A \vee B))$

| Mapping | $A \wedge B$ | $\neg(A \vee B)$ |
|---|---|---|
| 1 | Neuron (1, 1) | Neuron (1, 2) |
| 2 | Neuron (1, 1) | Neuron (1, 0) |

**Formula 39:** $\neg((A \wedge B) \vee \neg(A \vee B)) \wedge (A \vee B)$

| Mapping | $A \wedge B$ | $A \vee B$ | $\neg((A \wedge B) \vee \neg(A \vee B))$ | $\neg(A \vee B)$ |
|---|---|---|---|---|
| 1 | Neuron (1, 1) | Neuron (1, 2) | Neuron (2, 0) | Neuron (1, 0) |
| 2 | Neuron (1, 1) | Neuron (1, 2) | Neuron (2, 1) | Neuron (1, 0) |
| 3 | Neuron (1, 1) | Neuron (1, 0) | Neuron (2, 0) | Neuron (1, 2) |
| 4 | Neuron (1, 1) | Neuron (1, 0) | Neuron (2, 1) | Neuron (1, 2) |

**Formula 40:** $\neg(((A \wedge B) \vee \neg(A \vee B)) \vee \neg(A \vee B))$

| Mapping | $(A \wedge B) \vee \neg(A \vee B)$ | $A \wedge B$ | $\neg(A \vee B)$ (left) | $\neg(A \vee B)$ (left) |
|---|---|---|---|---|
| 1 | Neuron (2, 1) | Neuron (1, 1) | Neuron (1, 0) | Neuron (1, 2) |
| 2 | Neuron (2, 1) | Neuron (1, 1) | Neuron (1, 2) | Neuron (1, 0) |
| 3 | Neuron (2, 0) | Neuron (1, 1) | Neuron (1, 0) | Neuron (1, 2) |
| 4 | Neuron (2, 0) | Neuron (1, 1) | Neuron (1, 2) | Neuron (1, 0) |

Table 2: The list of valid minimal mappings for the example network. For each intermediate node of each formula, we specify which neuron corresponds to that node.

# D ADDITIONAL PLOTS

## D.1 TARGET GATE VARIATION

Figure 5 contains the total number of computational abstractions obtained after fixing $k = 3$ and $n = 1$ and sampling the target gate from the following list: AND, OR, XOR, IMP.

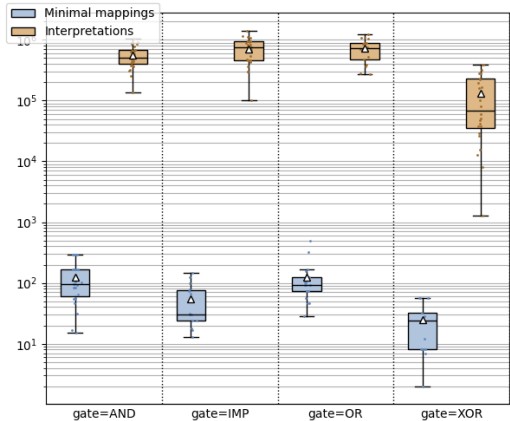

Figure 5: Total number of interpretations found in the circuit-first approach and of mappings found in the algorithm-first approach, grouped by target gate.

Figure 6 contains the results of the same experiment but displays separate plots for the number of circuits and interpretations per circuit (resp. algorithms and mappings per algorithm) for each network.

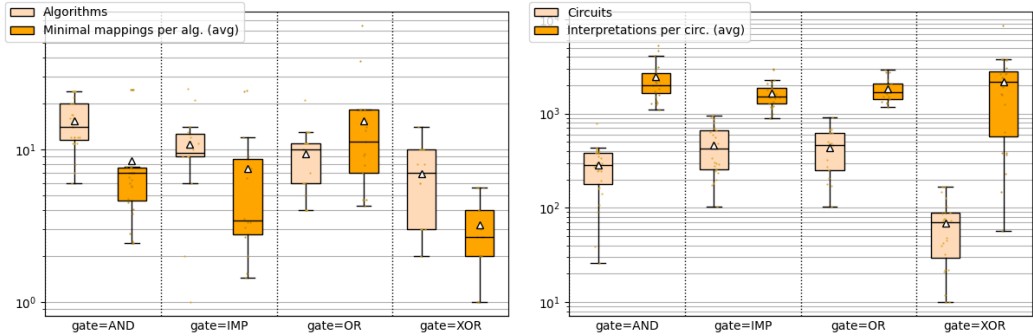

Figure 6: Left: Number of circuits and average interpretations per circuit found in the circuit-first approach. Right: Number of algorithms and average mappings per algorithm found in the algorithm-first approach (right). Results are grouped by target gate over 100 experiments.

## D.2 ARCHITECTURE SIZE

Figure 7 contains additional plots for the experiment described in 4.1.1, in which we vary the architecture size.

## D.3 MULTI-TASK TRAINING

We report in Figure 8 additional plots for the experiment in which we vary the number of gates the model is being trained on (described in 4.1.2).

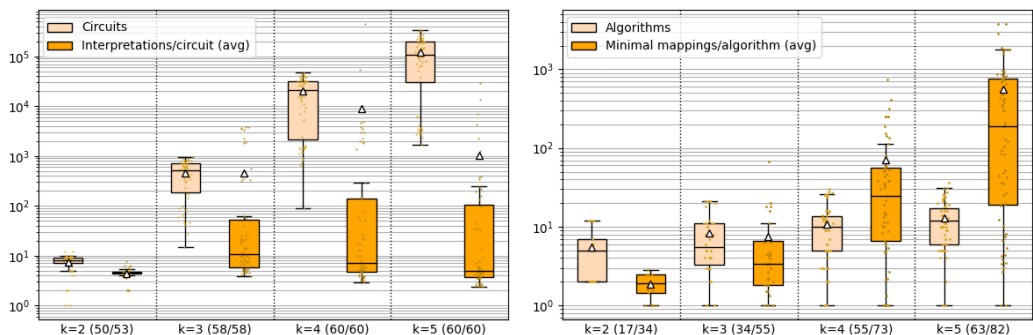

Figure 7: Number of abstractions found in the circuit-first approach (left) and the algorithm-first approach (right) as a function of the architecture size.

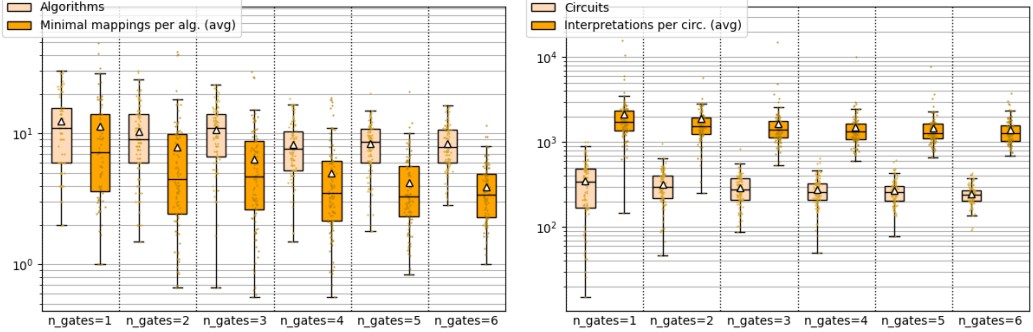

Figure 8: Number of abstractions found in the circuit-first approach (left) and the algorithm-first approach (right) as a function of the number of training tasks.

## D.4 LOSS CUTOFF

We report in Figure 9 plots for the experiment in which we apply the basic setup with $n = 1$ and $k3$ on a set of networks, trained while varying the loss cutoff from $10^{-1}$ to $10^{-6}$. For the algorithm-first approach, a two-sample t-test indicates a modest but significant decrease in the number of algorithms found when the loss cutoff is lower or equal to $10^{-5}$. In contrast, the number of mappings per algorithm does not statistically vary. For the circuit-first approach, significantly fewer circuits and interpretations per circuit are found when the loss cutoff is high (0.1), but values do not otherwise vary for lower loss values. In addition, we found that multiple computational abstractions can still be identified in randomly initialized networks that happen to implement a logic gate (i.e. without training).

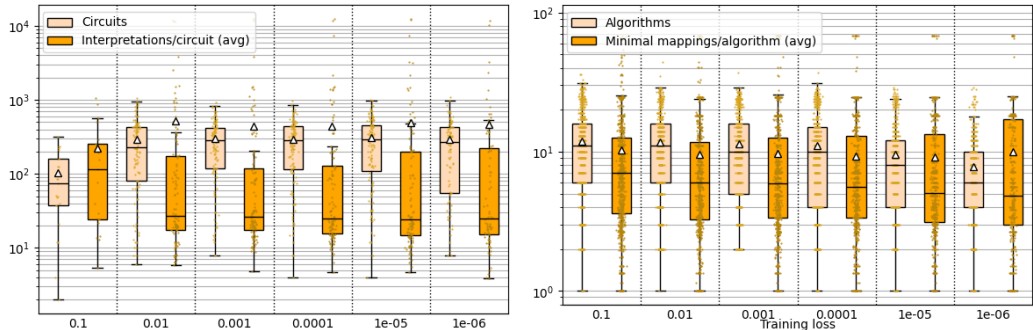

Figure 9: Number of abstractions found in the circuit-first approach (left) and the algorithm-first approach (right) as a function of the neural network's training loss cutoff.

## D.5 EFFECT OF NOISE

We report in Table 3 the number of abstractions found in the basic setup (without noisy inputs) for both approaches, and compare it to the results found when repeating the same setup while adding Gaussian noise from $\mathcal{N}(0, 0.1)$ to the inputs at training time. We also report the $p$-value obtained with Welch's t-test.

| Criterion | Basic setup | Noisy setup | $p$-value |
|---|---|---|---|
| Algorithms | 2.70 | 2.56 | 0.766 |
| Minimal mappings/algorithm (avg) | 2.84 | 3.05 | 0.617 |
| Total mappings | 9.86 | 9.89 | 0.959 |
| Circuits | 52.3 | 28.6 | **< 0.001** |
| Interpretations/circuit (avg) | 2,270 | 16,400 | **< 0.001** |
| Total interpretations | 107,000 | 285,000 | **< 0.001** |

Table 3: Number of abstractions found for both approaches in each setup (averaged over all networks).

## D.6 TRAINING DISTRIBUTION

The influence of the training distribution was investigated through the following procedure:

1. Sample a random neural network $NN$ and target gate with $n = 1$ and $k = 3$

2. Draw $x_1, \ldots, x_4$ from $U_{[0,1]}$

3. Train $NN$ on a skewed input distribution, with weights $\frac{x_i}{\sum_i x_i}$ for each input $i$, and a loss cutoff of $10^{-3}$.

4. Exhaustively enumerate all circuits, interpretations, algorithms, and mappings as in the basic setup.

5. Repeat from step 1.

We repeated those steps 100 times, resulting in varying training distributions with joint entropy varying from 0.8 to 2.0 bits. We then performed a linear regression of the resulting counts as a function of the distribution's joint entropy. We give in Table 4 the results for this experiment, which show that the results of the algorithm-first approach do not significantly depend on the training distribution. On the other hand, having an unbalanced distribution causes an increase in the number of circuits and total interpretations found in the circuit-first approach, but a decrease in the number of interpretations per circuit.

| Criterion | Slope | Intercept | $p$-value |
|---|---|---|---|
| Algorithms | -0.21 | 2.86 | 0.857 |
| Minimal mappings/algorithm (avg) | -0.14 | 1.42 | 0.807 |
| Total mappings | -0.17 | 8.82 | 0.974 |
| Circuits | -326 | 556 | **0.001** |
| Interpretations/circuit (avg) | 1,235 | 139 | **0.029** |
| Total interpretations | -174,000 | 367,000 | **0.018** |

Table 4: Linear regression performed on the number of computational abstractions as a function of the training distribution's joint entropy (in bits).

