# OpenReview forum: "Everything, Everywhere, All at Once: Is Mechanistic Interpretability Identifiable?"
_ICLR.cc/2025/Conference — ICLR 2025 Poster_

### Official Review · Reviewer_Q4uc · 2024-11-01

**Soundness:** 3
**Presentation:** 3
**Contribution:** 3
**Rating:** 6
**Confidence:** 2

**Summary:**

- The objective of the paper is to answer the question, do current criteria in mechanistic interpretability (MI) guarantee the identifiability of the explanation?
- Authors sort MI methods into two broad strategies:
    - _where-then-what_ focuses on finding a subset of the network – a circuit – that captures most of the information flow from in- to outputs. Once this circuit is identified, the next step is to interpret its components (features) to derive the explanatory algorithm.
    - _what-then-where_ starts by identifying candidate algorithms and then searches subspaces in the neural network where the algorithm may be implemented, using causal alignment between the explanatory algorithm’s states and the network’s internal states.
- They stress testing both methods with toy models: small MLPs trained on logic gates. They performed three main types of searches to test different interpretability criteria:
    - Circuits search: Looking for subnetworks that perfectly replicate the model's behavior
    - Interpretations search: Trying to map neurons to logical gates in a way that's consistent with their activations
    - Mappings search: Testing different ways to map logical gates to groups of neurons
- Key findings in toy models: they found multiple valid interpretations for the same network.
    - 85 different circuits that achieved perfect accuracy
    - An average of ~536 possible logic gate interpretations per circuit
    - 159 perfect minimal mappings between algorithms and neurons
    - In total, over 45,000 different possible computational abstractions
- Experiment larger NN trained on a subset of MNIST revealed similar dynamics
    - The circuit search found over 3000 valid circuits

**Strengths:**

- The authors introduce their own taxonomy and formalization (e.g. where-then-what/what-then-where, Circuits, Mappings, etc) for important concepts discussed in the paper. While I haven’t fully wrapped my head around the usefulness of the taxonomy, I appreciate the effort to deconfuse different interpretability methods. I think this is the strongest suit of the paper and I wish they had focused the paper on the taxonomy and less on the experiments.
- I enjoyed the writing style, and it was easy for me to follow, particularly Sections 2 and 3. I also found Figure 2 to be helpful. For the _what-then-where_/_where-then-what_ split.
- The paper touches on an important topic within interpretability – a lack of quality in the discourse around helpful metrics for interpretations.

**Weaknesses:**

- While I think the taxonomy could bear some relevant fruit, for experimentally testing the metrics using NN this small, it seems challenging to generate insights that are relevant to interpretability as a total. From my understanding, we are not close to having high-confidence circuit interpretations (“perfect circuits”) in the first place, so working under this assumption might be several steps ahead.
- Continuing on the larger NN experiments: I struggled to understand the points in line 479 ff. What is in this case the "valid circuit"? I hoped this section could have bridged a gap toward showing how this metric could be used in the future but failed to show its limitations in this instance. (But maybe I simply oversaw that.)
- Lastly, I think there is also too little emphasis on existing literature that clearly touches on the underlying problem: disentanglement. I would consider modeling the paper around the taxonomy and then focusing on existing research and problems in relation to known circuits, such as IOI.

**Questions:**

(From the weakness section)
- What is in the case of the MNIST NN the "valid circuit"?

---

> ### Author Response · Authors · 2024-11-20
> **Response to the reviewer**
>
> We would like to thank the reviewer for their comments and for appreciating our distinction between "what-then-where" and "where-then-what" approaches to mechanistic interpretability. Below, we address potential misunderstandings and clarify our contributions.
>
> > What is in the case of the MNIST NN the 'valid circuit'?
>
> In the MNIST experiment, like all our experiments, there is no concept of one single "valid circuit." The field of mechanistic interpretability proposes various criteria to define what constitutes an accepted explanation. Our experiments do not require the notion of a "true" or "valid" circuit, but only test whether the criteria induce a unique accepted explanation.
> Specifically, we show that the existing criteria suffer from identifiability issues, they lead to multiple incompatible circuits that are all equally plausible according to the criteria. It is important to note that we are not testing specific interpretability methods; rather, we are testing the criteria underlying those methods. Furthermore, we are not proposing new metrics in this work. Instead, we provide counterexamples illustrating that existing criteria fail to yield a unique solution (a unique explanation). This raises foundational questions about the assumptions and objectives of mechanistic interpretability that we discuss at length in Section 5.
>
> > It seems challenging to generate insights that are relevant to interpretability as a total.
>
> Our focus is on testing the interpretability criteria themselves. These criteria are meant to be applicable across contexts, including small neural networks. By focusing on small neural networks, we can enumerate all possible explanations exhaustively and demonstrate counterexamples that challenge current criteria. Such stress testing is intractable at scale, but our experiment with a larger NN on MNIST shows a lower bound on the number of circuits proving the problem remains at this scale.
> Our work is stress-testing the foundations of our field with respect to one property: identifiability. Our work seeks to contribute constructively to the field by questioning the underlying definitions of what is a valid explanation. A clearer understanding of what constitutes an explanation will ultimately guide the development of more reliable methods for generating them.
>
> > "Too little emphasis on existing literature that clearly touches on the underlying problem: disentanglement.
>
> Our work does not target specific methods for finding explanations or critique particular instances of circuits previously identified using these methods (like IOI). Instead, we address a more foundational question: whether the criteria used to evaluate explanations reliably induce a unique solution (i.e., a unique explanation). Through extensive empirical evidence, we demonstrate that current criteria fail in this regard. While disentanglement-related literature provides valuable insights, our primary aim is to stress-test the criteria themselves, not to evaluate or propose methods based on disentanglement.
>
> Finally, the source code used for our experiments will be released upon publication.
>
> We hope this response clarifies our contributions and addresses any misunderstandings. Our work is intended as a constructive step toward more rigorous definitions and criteria for explanations, which we believe will ultimately strengthen the field of mechanistic interpretability. Please let us know if we can further clarify our contributions.

---

> > ### Author Response · Authors · 2024-11-25
> >
> > Dear Reviewer,
> >
> > With the author-reviewer discussion period ending soon, we would like to send a gentle reminder that we have submitted the rebuttal to address the given comments. We sincerely appreciate your feedback and are happy to address any additional questions you may have during this discussion period.
> >
> > We thank you again for taking the time to review our work.

---

### Official Review · Reviewer_S8oD · 2024-11-01

**Soundness:** 2
**Presentation:** 3
**Contribution:** 3
**Rating:** 6
**Confidence:** 4

**Summary:**

The authors investigate the potential issue of identifiability in mechanistic interpretability through experiemnts on small MLPs where (isolated) circuits are ennumerated and assessed fairly exahustively. They find identifiability is an issue at all levels: in the number of subcircuits functionally aligned with the full network, in the number of algorithms consistent with the behavior, and in the mappings between algorithms and circuits. This problem gets worse as architecture size increases, and training on a greater number of tasks only mitigates this issue to some extent.

**Strengths:**

* Well written
* Well referenced
* Adresses an issue of interest to the interpretability community
* Provides exhaustive experiments with well-understood ground truth.
* Investigates the effects of various variables (architecture size, number of tasks, noise) on the identifiability problem.

This kind of study is very much needed.

**Weaknesses:**

I first summarize the most important points and then elaborate.

* It is unclear whether the conclusions about identifiability problems can be stated generally or only in particular for circuits as isolated objects (circuit discovery through zero-ablation), which might mischaracterize the functioning of the network as a whole.
* The possible improvements advertised in the abstract/introduction are rather only sketched in section 5.1.


Is the large number of circuits/possible explanations due to looking for circuits in isolation (e.g., via zero ablation) rather than working in context with the rest of the network (e.g., via activation patching)?

Line 320 describes the circuit isolation procedure. This is equivalent to zero-ablation and the criterion is equivalent to the definition of suficcient circuit.
How would identifiability look like if we chose to define circuits as they function in the context of the full network? See for example the definition of circuit via activation patching in Adolfi et al., 2024.

Isn’t it possible that many of the isolated circuits discovered through zero-ablation are mischaracterizations of the in-context functioning of the circuits as they are embedded in the full network?


Line 080: “a model’s behavior should have a single, well-defined explanation”. There is no citation here and it is unclear where this intuition is coming from, what is it’s theoretical support, etc. To offer a counter-intuition: consider a circuit that is sufficient on its own to mimic the behavior of the full network over some input domain; such a circuit need not be unique. Trivially, the circuit plus additional neurons to form the full network is another such circuit. But there is no contradiction in intuiting that multiple such circuits of different sizes, with partial or no overlap exist in the network and, in principle, offer alternative (perhaps incompatible?) ‘explanations’ (see Adolfi et al. 2024 for theoretical analyses).

On Line 091: the authors mention “the identifiability properties of current MI criteria”. The criteria of interest that define circuits leave open the possibility that these circuits are not unique. So the definition of these circuits does not preclude non-identifiability unless the uniqueness property is trivially appended to the definition. This leads one to suppose that uniqueness under the typical definition of circuits is a property left to be determined empirically. It could, in principle, be motivated theoretically, but I see nothing in that direction here. Is it possible to provide some theoretical motivation for uniqueness that is not trivially stipulated but justified from first principles?

If a network implements the same functionality for different inputs through different circuits and algorithms, does this really make mechanistic interpretation hopeless? (i.e., in this case, is only a functional explanation capable of unifying all the existing ‘explanations’?). It would be useful to have any assumptions about satisfactory explanations made explicit in the manuscript.

Line 489: “the challenge lies in defining criteria that distinguish valid explanations from misleading ones.”
It seems to me that, conceptually, identifiability does not pose a problem for distinguishing misleading from valid explanations. The problem arises only if an explanation is presented as unique or valid for a full input domain when this is not so. This issue might warrant some clarification.

Line 490: “According to MI, the explanatory algorithm should be unique, meaning multiple competing explanations should not exist.” But this statement is made without citation. This assumption seems ill-founded to begin with, for the reasons mentioned above. Where does the criterion come from?

**Questions:**

Minor comments, questions, and suggestions:

Line 403 states that “larger architecture…could also lead to greater overparameterization”. This could benefit from elaboration; in particular, how larger architecture could lead to a reduction in the number of valid abstractions.

On Line 065, “Given the near impossibility of exhaustively searching all possible algorithms across all subsets of a neural network”, I might suggest to reframe this not as impossibility but as intractability, infeasibility or implausibility. Certain interpretability queries might have large search spaces that could nevertheless be searched efficiently. The relevant property is the complexity of the interpretability query, not merely the size of the search space. For computational complexity analyses of circuit discovery, see Adolfi et al 2024.

On Line 067, the authors state “researchers have developed approximation methods with different assumptions and trade-offs”. It seems to me that the circuit discovery methods that are typically developed are heuristics for circuit finding, not approximation algorithms with any proven guarantees. In any, case it would be useful if the authors can distinguish between these two categories in their descriptions.

Citation to Van Rooij on Line 046 does not seem to fit with the corresponding sentence, as that paper does not deal at all with interpretability, as opposed to Lindsay, 2024, which is indeed an approapriate citation. For examples of studying the fundamental properties of (inner) interpretability queries see Adolfi et al., 2024.

Section 2.1 mentions interpretability work on transformer models but only in language. An example from vision transformers can be found in Vilas et al. 2023.

Please clarify the notation in Definition 4.

Line 229 makes an implicit statement about computational complexity but provides no citation. See Adolfi et al. 2024 for relevant complexity analyses. This is also relevant to the statement on Line 257. Here it would also be useful to clarify how uniform random sampling “approximates” the desired measure, as this seems non-obvious. Perhaps the authors mean random sampling is a heuristic with unknown properties?

Line 494 states that current MI methods can only approximate their targets because exhaustive enumeration is impossible for large models. This is technically incorrect, as even for some problems with exponential serach spaces, efficient search algorithms that find optimal solutions are possible. The relevant notion is the computational complexity of the interpretabililty queries, not simply the size of their search space (see Adolfi et al., 2024).

Section 2.1 describes a parallel between AI interpretability and neuroscience. A framework that draws from lessons grounded in this parallel is described in Vilas et al. 2024. This framework provides a nice embedding for the what-where distinction, corresponding to the algorithmic and implementational levels, respectively.

The problem of identifiability interacts in interesting ways with the computational complexity of circuit finding. Adolfi et al. 2024 analyses circuit queries that are relevant to the authors’ points on identifiability. See, for instance, counting problems which ask for the number of circuits in a train neural network that have a certain property (e.g., they are sufficient for a behavior). Furthermore, if the number of sufficient circuits is typically large, heuristics for otherwise intractable problems (e.g., sufficient circuit) could seemingly find their targets in a feasible amount of time. In this scenario, non-identifiability is an important catch.

---

> ### Author Response · Authors · 2024-11-20
> **Response to the reviewer**
>
> We greatly thank the reviewer for their extensive and insightful feedback, as well as for appreciating the usefulness and relevance of our work and the exhaustiveness of our experiments. This review has been highly instrumental in the revisions we made to the paper.
>
> We first give a high-level response to the most important points raised:
>
> __On whether the conclusions about identifiability problems can be stated generally or only in particular for circuits as isolated objects:__
> It may indeed be the case that identifiability issues of the "where-then-what" scenario stem from the isolation of circuits from the rest of the network. This would point to a problem with the definitions, in this work, we use the precise goal of circuit discovery that has been formulated in previous work. Also, we emphasize that in the what-then-where, the IIA requirement is a strong case of activation patching. It relies on exhaustively applying counterfactual interventions to parts of the network in context. Specifically, the computation flow goes through the entirety of the network's components when evaluating mappings. As a result, the incompatible explanations found in this strategy indicate that identifiability problems cannot simply be reduced to mischaracterization of the functioning of the network as a whole.
>
> __On the extension of the possible improvements mentioned in the abstract/introduction:__
> We have largely rewritten Section 5 (discussion). Specifically, we now give in Section 5.2 several avenues of research through which identifiability issues may be resolved. Amongst these, we now discuss the inner interpretability framework mentioned in Vilas et al. (2024) as a promising path toward strengthening MI as an experimental science.
>
>
> We now proceed to respond to individual feedback points:
>
> __On why a model's behavior should have a single, well-defined explanation, and whether MI assumes unicity:__
> We have introduced a new Section (2.4), in which we now argue that the unicity of explanation of a phenomenon is a strong intuition deeply rooted in human reasoning. After slightly rephrasing the abstract and introduction, we have removed phrasing implying that unicity is an explicit requirement. We now frame the unicity of explanation as a property that we might intuitively expect but that is clearly violated in our experiments. We also argue by extracting quotes (documented in Appendix C) that previous MI research implicitly assumes a unique explanation.
>
> > Trivially, the circuit plus additional neurons to form the full network is another such circuit.
>
> We exclude such a scenario: every time we find two circuits that perfectly compute the behavior of the full network but one is included in the other, we only keep the smallest one. Similarly, for the what-then-where approach, we also only keep the smallest mapping that we find. That's why we always report the number of "minimal mappings" and not the number of mappings.
>
> __On whether non-identifiability makes MI hopeless:__
> We have clarified our positioning towards the need for identifiability. In Section 5, we question whether the lack of unicity poses a problem (5.1) and whether it is achievable in the first place (5.3). In summary, we think that, as a community, we could: (i) clarify the epistemic goals of an explanation (potentially accepting the lack of unicity), (ii) investigate less permissive criteria, and (iii) embrace broader frameworks like the inner interpretability framework.
>
> __On “the challenge lies in defining criteria that distinguish valid explanations from misleading ones”:__
> We have removed this line from the manuscript, as it did not accurately reflect our positioning.
>
> In addition, we have applied most of the suggested minor changes, including the following:
> - Stronger justifications as to why a larger architecture could also lead to greater overparameterization have been given (l. 398-401).
> - The “near impossibility of exhaustively searching all possible algorithms” has been rephrased as "intractability" (l. 49).
> - The distinction between circuit-finding heuristics and approximation algorithms has been clarified (l. 53-57).
> - We have edited several citations: removal of the Van Rooij citation, and addition of Vilas et al., 2023 on vision transformers in MI, Adolfi et al., 2024 on the inner interpretability framework, and Adolfi et al., 2024 on computational complexity.
> - We have clarified the notation in Definition 4 (l. 197).
> - The initial statement about computational complexity has been adapted and sourced (l. 204)
>
> Finally, the source code used for our experiments will be released upon publication.
>
> Please let us know if we can provide any additional information to clarify our work.

---

> > ### Comment · Reviewer_S8oD · 2024-11-22
> > **Response to rebuttal**
> >
> > Thanks for the clarifications/changes. I have no further comments at this point except this:
> >
> > The intractability I was referring to above has to do with the complexity status of interpretability queries (the intrinsic hardness of certain computational problems that formalize circuit discovery), not to the (more obvious) time complexity of enumerating an exponential number of circuits. The statements in the manuscript about “intractability of enumerating” should probably be amended/supplemented to reflect this, since the more relevant notion here is the intractability of problems, not the complexity of enumerating (which is merely one possible approach to problems).
> >
> > For reference, see e.g.:
> >
> > https://www.cs.uwyo.edu/~jhitchco/papers/npcmcsp.pdf
> >
> > https://doi.org/10.48550/arXiv.2410.08025

---

> > > ### Author Response · Authors · 2024-11-25
> > >
> > > Thank you for your positive assessment of our answer.
> > >
> > > In line 49, our usage of "intractable" refers to the time complexity of enumerating all possible candidate circuits and it is its intended usage.
> > >
> > > However, we mention the fundamental intractability discussed by [1] and [2] about the intrinsic computational hardness of interpretability queries in section 5.2 regarding the possibility that MI is fundamentally underdetermined. We have updated the sentence citing these articles to better emphasize this aspect, it is now:
> > > "The intrinsic computational hardness of interpretability queries (Adolfi et al., 2024a;b) suggests that MI may have fundamental limits, leaving it possibly underdetermined."
> > >
> > > Thank you again for the constructive discussion. Let us know if there are additional ways in which we can further improve the paper.

---

> > > > ### Comment · Reviewer_S8oD · 2024-11-25
> > > > **No further comments**
> > > >
> > > > Thanks. No further comments at this point.

---

> ### Author Response · Authors · 2024-11-28
>
> Dear Reviewer,
>
> We would like to express our gratitude for the thoughtful and thorough review of our manuscript. The detailed feedback has been valuable in improving the overall quality of the paper. We also appreciate your suggestions regarding computational complexity and related references.
>
> We are confident that the revisions we have made reflect your input and contribute to a clearer and more robust manuscript. If we have addressed your concerns, we would be grateful if you might consider revisiting your score.
>
> Thank you once again for your constructive feedback.

---

### Official Review · Reviewer_J9MJ · 2024-11-02

**Soundness:** 3
**Presentation:** 3
**Contribution:** 4
**Rating:** 8
**Confidence:** 5

**Summary:**

This paper investigates the \emph{identifiability} of mechanistic explanations from interventional evaluations on toy neural networks. The authors find clear non-identifiability at multiple stages of the interpretability pipeline: multiple interpretations can exist for a circuit, multiple circuits can generate the same behavior, each algorithm can be aligned to multiple activation subspaces, etc.
This non-identifiability persists regardless of whether MI explanations are generated first by localizing a subset of the network, then deriving an interpretation, or first generating a candidate algorithm and trying to then find an activation subspace corresponding to that algorithm.

**Strengths:**

A rare case of compelling and relevant deconfusion experiments which doesn’t devolve into over-claims. The authors do well in the discussion highlighting that identifiability may not be needed for all applications of mech interp, and that the non-identifiability observed here is on toy-models, so might not extend to larger models trained on multiple tasks.

The “what-then-where” strategy implemented here appears to solely utilize an approach based on Distributed Alignment Search (DAS), which makes the results about non-identifiability quite relevant to recent discussion:

Makelov, A., Lange, G., & Nanda, N. (2023). Is this the subspace you are looking for? An interpretability illusion for subspace activation patching. arXiv. https://arxiv.org/abs/2311.17030

Wu, Z., Geiger, A., Huang, J., Arora, A., Icard, T., Potts, C., & Goodman, N. D. (2024). A reply to Makelov et al. (2023)'s "Interpretability Illusion" arguments. arXiv. https://arxiv.org/abs/2401.12631

**Weaknesses:**

Primary weakness: The most novel contribution of this paper are its experimental results, which don’t have enough description: Appendix B only contains aggregated results about the number of circuits and average interpretations per circuit found, but e.g. lacks examples of said circuits for qualitative validation. This significantly undercuts my ability to validate the correctness of the experiments.

In addition, the authors acknowledge appropriately that identifiability may not be necessary if the goal of MI is merely to steer a model. However, much MI work is driven by a desire to simply further scientific understanding of language models. What types of scientific inquiries require computational identifiability, and which do not? The paper could be strengthened by further discussing how much identifiability matters if the goal is scientific understanding, rather than just the model steering mentioned in lines 520-527.

**Questions:**

1. Could you elaborate on the meaning of “incompatible” in line 355? The paper would benefit from a clear example of two incompatible explanations, ideally in the main body.

2. Please include in the appendix random examples of some of the circuits found so that they can be qualitatively assessed by readers.

3.Could you comment on how these results relate to the discussion of Makelov et al. (2023) and Wu et al. (2024), cited above in section “Strengths”?

---

> ### Author Response · Authors · 2024-11-20
> **Response to the reviewer**
>
> We greatly thank the reviewer for their time and feedback, and for highlighting the relevance of our work and the discussion on the limits of its scope.
>
> To address the lack of examples, we have added a new appendix B, which contains more examples of circuits, interpretations, algorithms, and mappings found. In addition, we have clarified the text in the article to emphasize the fact that the example circuits given in Figure 2 were also found as part of our experiments. We hope this will help readers visualize and assess the quality of our work.
>
> We have clarified our position concerning identifiability in two ways; first, we now argue in section 2.4 that the need for identifiability stems from a strong intuition rooted in human psychology and that while not clearly stated, it is already an implicit assumption of MI. The paper now frames the unicity of explanation as a property that we might expect but do not find.
> Second, we have extensively rewritten the discussion (Section 5), in which we describe in which scenarios the lack of unicity may or may not matter. In this context, we now cite the recent debate about the interpretability illusion as an example of why explicitly stating the purpose of the explanation is relevant to the question of identifiability.
>
> In addition, we would like to clarify that the what-then-where approach in our work does not rely on DAS. While DAS only explores a part of the search space via gradient descent and therefore produces an approximate solution, in our experiments we find all exact maximizers of IIA by exhaustive enumeration. Furthermore, we compute the true IIA value, while DAS typically only approaches it by randomly sampling inputs.
>
> Finally, the source code used for our experiments will be released upon publication.
>
> Please let us know if we can provide any additional information to clarify our work and thank you again for the feedback.

---

> > ### Comment · Reviewer_J9MJ · 2024-11-22
> > **Good responses, maintaining score**
> >
> > Thank you for clarifying that the what-then-where approach does not rely on DAS. Admittedly, as I'd reflected over the last few weeks I'd been troubled by the apparent reliance on DAS, as activation patching can trivially score high on the IIA if enough dimensions are patched. I was going to lower my score accordingly, but your rebuttal helped clarify that this method does not rely actually rely on patching.
> >
> > I will maintain my score.

---

> ### Author Response · Authors · 2024-11-28
>
> Dear Reviewer,
>
> We sincerely thank you for your thoughtful and careful consideration of our work, as well as for maintaining the score. We appreciate the time spent reflecting on the details, and we're glad that our clarifications helped address the concerns about the reliance on DAS. Should any further questions arise, we are happy to provide additional insights.
>
> Thank you again for your time and contribution.

---

### Official Review · Reviewer_ExBt · 2024-11-03

**Soundness:** 3
**Presentation:** 4
**Contribution:** 3
**Rating:** 8
**Confidence:** 3

**Summary:**

A summary of mechanistic interpretability is provided. Two approaches are proposed that attempt to interpret a “simpler” algorithm that emulates the behavior of a trained neural network in terms of a circuit. One approach focuses on modelling the behavior of the full network before finding a subset of related nodes, while the second approach focuses on finding an “important” sub-network of the full network, whose behavior is then interpreted. Both approaches are showcased in simple toy examples.

**Strengths:**

The manuscript is clearly written, with a good explanation of the context the work is placed in. The toy examples provided are illustrative. The overarching conclusion is that uniqueness of an explanation should generally not be expected within the context of mechanistic interpretability, and while a similar analysis cannot be conducted on large-scale models, it is likely that the same behavior could be expected. This insight is important in many practical applications where network interpretations are required (post-training) by a practitioner.

**Weaknesses:**

The issue of uniqueness of an explanation is addressed in the context of mechanistic interpretability. However, the “incompatibility” of different explanations is not substantially addressed. A more formal framework in which incompatibility can be “measured” would be very interesting, along with analyzing questions on differentiating between equivalence classes of explanations.

**Questions:**

If the training set is drawn from a distribution with certain biases, there may be correlations that essentially encourage multiple “conflicting” interpretations of a network. Can we resolve some of the issues that arise by putting conditions on the training distributions?

What would (be expected to) happen if, in a simple toy example, an experiment was repeated with a perfect training error, with or without overfitting? Would we see a qualitatively different distribution of explanations?

---

> ### Author Response · Authors · 2024-11-20
> **Reponse to the reviewer**
>
> We sincerely thank the reviewer for their time, insightful comments, and kind remarks about the clarity and interest of our work.
>
> __On the definition and measurement of “incompatibility”:__
> We have expanded the definition of incompatibility in the manuscript (section 2.4). We broadly define fully incompatible explanations as pairs of explanations that share the same epistemological goal but lack overlap in location (within the neural network, the "where") and/or differ in internal states (e.g., algorithms, the "what").
> We agree that compatibility can also be partial, but this is out of the scope of this work and we leave open formal definitions and measurements of compatibility for future research.
> Our study primarily focuses on at least partially incompatible explanations (i.e., no full overlap). The number of incompatible explanations that we find is particularly large and examples of fully incompatible explanations have been added in Appendix B.
>
> __On the distribution of the training set:__
> We have added an experiment where the training procedure is repeated with varying training distributions. The results, in Appendix C.5, suggest that training biases do not significantly affect the conclusion.
>
> __On training error and overfitting:__
> We have expanded our previous experiment on training dynamics, in which we analyzed scenarios where training was stopped at different loss cutoffs. The manuscript now includes additional figures to that extent (Appendix C.4). Our findings indicate that approaching perfect training error does not substantially alter the number of explanations found. Interestingly, many incompatible explanations can also be identified in randomly initialized (untrained) networks that just happen to implement the target logic gate.
>
> Finally, the source code used for our experiments will be released upon publication.
>
> Please let us know if we can provide any additional information to clarify our work.

---

> > ### Author Response · Authors · 2024-11-25
> >
> > Dear Reviewer,
> >
> > With the author-reviewer discussion period ending soon, we would like to send a gentle reminder that we have submitted the rebuttal to address the given comments. We sincerely appreciate your feedback and are happy to address any additional questions you may have during this discussion period.
> >
> > We thank you again for taking the time to review our work.

---

### Meta-Review · Area_Chair_fr7j · 2024-12-18

**Metareview:**

This paper aims to answer the question of whether mechanistic interpretability guarantee identifiability through small-scale MLP network which allow tractable exhaustive enumeration in the experiments, and the finding is no. The reviewers are generally satisfied with the authors' response and revise draft and raised scores after rebuttal discussion accordingly. Hence, an acceptance is recommended.

Nevertheless, one part that is still remain unclear is the experiments in sec 4.3, which is rather short and need more clarification on what explanations/functionality are found in the mnist network. The authors are urged to update sec 4.3 with more details like in Appendix B.

**Additional Comments On Reviewer Discussion:**

Reviewers have various concerns including:
1. What's the motivation of requiring identifiability in MI?
2. The details of major experiments are lacking
3. Applicability to large network in practice.

The authors answered 1 and 2 well by revising the draft with more accurate descriptions and providing details in the appendix. For 3, I think the authors can provide more results on sec 4.3 which will be beneficial to the community.

---

### Decision · Program_Chairs · 2025-01-22

Accept (Poster)